# UBC9-Mediated SUMO Pathway Drives Prohibitin-1 Nuclear Accumulation and PITX1 Repression in Primary Osteoarthritis

**DOI:** 10.3390/ijms26136281

**Published:** 2025-06-29

**Authors:** Roxanne Doucet, Abdellatif Elseoudi, Bita Rostami-Afshari, Mohamed Elbakry, Maryam Taheri, Martin Pellicelli, Cynthia Picard, Jean-François Lavoie, Da Shen Wang, Patrick Lavigne, Kristen F. Gorman, Wesam Elremaly, Alain Moreau

**Affiliations:** 1Viscogliosi Laboratory in Molecular Genetics of Musculoskeletal Diseases, Azrieli Research Center, CHU Sainte-Justine, Montreal, QC H3T 1C5, Canada; 2Program of Molecular Biology, Faculty of Medicine, Université de Montréal, Montreal, QC H3T 1J4, Canada; 3Department of Biochemistry and Molecular Medicine, Faculty of Medicine, Université de Montréal, Montreal, QC H3T 1J4, Canada; 4Biochemistry Division, Chemistry Department, Faculty of Science, Tanta University, Tanta 31527, Gharbia Governorate, Egypt; 5Department of Surgery, Faculty of Medicine, Université de Montréal, Montreal, QC H3T 1J4, Canada; 6Orthopedic Division, Maisonneuve-Rosemont Hospital, Montreal, QC H1T 2M4, Canada; 7Department of Stomatology, Faculty of Dentistry, Université de Montréal, Montreal, QC H3T 1J4, Canada

**Keywords:** osteoarthritis, Prohibitin-1 (PHB1), PITX1, SUMOylation, UBC9

## Abstract

Osteoarthritis (OA) is a prevalent and debilitating joint disease in older adults with a complex etiology. We investigated the role of SUMOylation, a post-translational modification, in OA pathogenesis, focusing on the mitochondrial chaperone Prohibitin (PHB1) and the cartilage homeostasis transcription factor PITX1. We hypothesized that oxidative stress-induced SUMOylation promotes PHB1 nuclear accumulation, leading to *PITX1* downregulation and contributing to OA development. Analysis of cartilage specimens from 27 OA patients and 4 healthy controls revealed an increased nuclear accumulation of PHB1 in OA chondrocytes, accompanied by elevated levels of SUMO-1 and SUMO-2/3. Mechanistically, nuclear PHB1 interacted indirectly with SUMO-1 through a SUMO-interacting motif (SIM), and the deletion of this SIM prevented PHB1 nuclear trapping in OA cells. Furthermore, the SUMO-conjugating enzyme E2 (UBC9) encoded by the *UBE2I* gene was upregulated in knee OA cartilage, and its overexpression in vitro enhanced PHB1 nuclear accumulation. Consistently, transgenic mice overexpressing the *Ube2i* gene exhibited increased UBC9 in their knee cartilage, resulting in *Pitx1* downregulation and the emergence of an early OA-like phenotype in articular chondrocytes. Our findings uncover a novel role for UBC9-mediated SUMOylation in primary knee and hip OA. This pathway enhances PHB1 nuclear accumulation, contributing to *PITX1* repression and subsequent OA development. These results underscore the importance of SUMOylation in OA pathogenesis and suggest potential molecular targets for early diagnosis and therapeutic intervention.

## 1. Introduction

Osteoarthritis (OA) is a highly prevalent and disabling joint disorder that affects millions globally and arises from a multifactorial interplay among genetic predisposition, aging, biomechanical stress, metabolic alterations, and low-grade inflammation [1]. A central hallmark of OA pathogenesis is the progressive disruption of the delicate equilibrium between anabolic (cartilage matrix synthesis) and catabolic (cartilage matrix degradation) activities within articular chondrocytes [2]. Vidal-Bralo, L. et al. highlights the significant contributions of epigenetic modifications [3] and post-translational modifications [4,5], particularly those triggered by oxidative stress [6], in the dysregulation of chondrocyte function and cartilage homeostasis [7].

Mitochondrial dysfunction and the associated increase in oxidative stress, both recognized hallmarks of aging, have emerged as key players in OA initiation and progression. Articular chondrocytes, unlike many other cell types, are adapted to relatively hypoxic environments. However, as cartilage ages and degenerates, these cells become particularly vulnerable to insults that compromise mitochondrial integrity and function. Mitochondria, the primary sites of cellular energy production and key regulators of cellular homeostasis, are particularly vulnerable to oxidative damage. Mitochondria are also the primary sites of reactive oxygen species (ROS) production, and under conditions of stress, an imbalance between ROS generation and antioxidant defense leads to chronic oxidative stress. In the context of primary OA, dysfunctional mitochondria produce ROS, which, in turn, causes oxidative damage to cellular macromolecules, including DNA, protein, and lipids within chondrocytes [8,9]. This chronic oxidative stress directly contributes to chondrocyte senescence, apoptosis, and the proteolytic degradation of the extracellular matrix, thereby accelerating cartilage breakdown. This creates a vicious cycle where damaged cartilage further exacerbates oxidative stress, driving OA progression [8,9]. Intriguingly, within the mitochondria’s cellular machinery reside prohibitins (PHBs), a conserved family of mitochondrial proteins that are essential for maintaining mitochondrial integrity and function. Notably, Prohibitin-1 (PHB1) has been shown to exhibit diverse subcellular localization, shuttling between the nucleus, mitochondria, and cytosol and even associating with certain cell membrane receptors. Such diverse localization suggests a multifaceted role for PHB1 in cellular processes beyond its established mitochondrial functions [10,11,12,13,14,15,16,17,18].

Our previous work identified a critical link between PHB1 and primary knee/hip OA pathogenesis. We found that the nuclear accumulation of PHB1 in articular chondrocytes coincides with the onset of primary OA, leading to the transcriptional repression of paired-like homeodomain 1 (*PITX1*), a gene crucial for cartilage development and maintenance [19]. Remarkably, other studies have reported that PHB1 nuclear localization is associated with a senescent cellular phenotype, a hallmark of aging and potentially a contributing factor to OA pathogenesis [20,21,22]. Furthermore, PHB1 has been shown to interact with key transcriptional regulators such as retinoblastoma protein (RB), E2F transcription factor 1 (E2F1), and tumor protein p53 (p53), leading to the downregulation of E2F target genes, including *PITX1* [23].

To elucidate the mechanisms underlying primary knee and hip OA progression, we sought to unravel the underlying molecular mechanisms driving the aberrant nuclear accumulation of PHB1 in the context of OA. Recent evidence suggests that SUMOylation, a reversible post-translational modification [24] involving the covalent attachment of Small Ubiquitin-like Modifier (SUMO) proteins, may be a key factor. SUMOylation is a dynamic process that regulates protein localization, activity, and interactions [24,25,26,27,28,29,30,31,32,33,34]. The dysregulation of SUMOylation pathways has been implicated in various diseases, including aging and mitochondrial dysfunction [35,36]. Interestingly, SUMOylation machinery is closely linked to Promyelocytic Leukemia (PML) Nuclear Bodies (NBs) [PML-NBs], as NB formation primarily mirrors cellular oxidative stress, explaining NB prevalence in diseased but not in normal tissues. These findings imply that PML recruits UBC9 (Ubiquitin-Conjugating Enzyme E2 I), the central component of this machinery, to form NBs [37]. It facilitates the SUMOylation of many critical proteins by enhancing their sequestration into NBs, which serve as scaffolds [38]. Although the precise mechanisms by which SUMO modifications regulate diverse biological functions are still under active investigation, the deregulation of SUMOylation pathways is increasingly recognized to have significant and often drastic effects, contributing to the pathogenesis of several human diseases [39].

In this study, we hypothesized that increased SUMOylation activity, potentially involving specific SUMO isoforms and UBC9, drives PHB1 nuclear accumulation in OA articular chondrocytes. We investigated this mechanism, including the role of a PHB SUMO-Interacting Motif (SIM), and used in vitro and in vivo models to assess the impact on *PITX1*/*Pitx1* expression and the OA-like phenotype. Identifying SUMOylation as a driver of PHB1 nuclear localization and *PITX1*/*Pitx1* downregulation offers the potential for novel diagnostic and therapeutic strategies for OA.

## 2. Results

### 2.1. SUMO-1 Proteins Accumulate in the Nuclei of OA Articular Chondrocytes and Co-Localize with PHB1

The intracellular localization of PHB1, SUMO-1, and SUMO-2/3 in OA patients was performed by the immunofluorescence staining of human articular chondrocytes in 27 OA patients and 4 control subjects with corresponding antibodies. By using anti-PHB1 and anti-SUMO1 antibodies, the staining revealed that both proteins were co-localized in the nuclei of the articular chondrocytes of OA patients (Figure 1a). In contrast, in control subjects, SUMO-1 was predominantly located in the cytosol. However, articular chondrocyte staining with antibodies against SUMO-2/3 and PHB1 in the OA and control subjects did not show any nuclear co-localization, although both proteins were accumulated in the nuclei (Figure 1b).

Given that PML-NBs are strongly associated with the SUMOylation process [40,41,42,43], which is an essential part of PML-NB function during both nuclear body assembly and PML degradation, we performed co-localization experiments with antibodies against PML and SUMO proteins. Our results showed that either SUMO-1 (Figure 2a) or SUMO-2/3 (Figure 2b) were primarily localized in the PML-NBs of the OA chondrocyte nuclei compared to the control ones.

To explore the precise location of PHB1 in the nuclei of OA and control chondrocytes, immunostaining with antibodies against PHB1 or PML proteins was performed and then analyzed by confocal microscopy. The staining showed the nuclear co-localization of PHB1 in PML-NBs (Figure 3).

### 2.2. Is PHB1 Directly SUMOylated or Does It Interact with SUMOylated Partners in Primary OA?

To determine whether PHB1 is directly SUMOylated or interacts with SUMOylated partners, we first conducted an in silico analysis to identify putative SUMOylation sites or SUMO-interacting motifs (SIMs) within the human PHB1 protein. This analysis aimed to clarify the relationship between PHB1 and SUMO-1 and assess whether the nuclear trapping of PHB1 is mediated by SUMOylation. Accordingly, and as shown by a previous study, putative SUMOylation sites and one SIM were identified (Figure 4). The best-characterized canonical SBMs/SIMs are linear motifs, which are all characterized by a stretch of hydrophobic residues with a loosely conserved consensus sequence ([V/I]-x-[V/I]-[V/I] or [V/I]-[V/I]-x-[V/I]) [44,45]. Based on this consensus motif, we observed only one SIM, which followed the sequence between the amino acids at position 76 and 80 (Figure 4).

Because numerous SUMOylation sites are cryptic and do not necessarily match consensus ones, we decided to address the question by performing a classical in vitro SUMOylation assay using a purified GST-PHB1 protein, with GST and GST-RanGap1 as negative and positive SUMOylation controls, respectively. Consequently, we did not find evidence that PHB1 is directly SUMOylated by SUMO-1 (Figure 5). It has been reported that SIM can facilitate the SUMOylation of proteins [46].

To investigate the non-covalent interaction between PHB1 and SUMO-1, we generated several PHB1 constructs: one lacking the identified consensus SUMO-interacting motif (PHB1_ΔSIM) and another in which the C-terminal nuclear export signal (NES) was deleted and replaced with a nuclear localization signal (PHB1_NLS; Figure 6a). By using a co-immunoprecipitation assay with nuclear fractions of U2OS cells that overexpressed PHB1 or PHB1-ΔSIM, we conclusively demonstrated that PHB1 interacted with SUMO-1 through its SIM given that the deletion of SIM abrogated this interaction (Figure 6b). Moreover, those different plasmids were transfected in the C28/I2 human chondrocytes cell line and the protein expression was determined by Western immunoblotting (Figure 6c). The deletion of the SIM in PHB1 almost completely abolished its nuclear accumulation in transfected cells.

### 2.3. UBC9-Mediated SUMOylation Promotes Nuclear Accumulation of PHB1 in Primary OA

To identify the source of increased SUMOylation in primary knee OA, we initially investigated the contribution of UBC9, since this enzyme is the unique E2 ligase involved in the SUMOylation pathway. We used an immunohistochemical assay to compare UBC9 in normal and OA articular chondrocytes. Indeed, we observed that UBC9 protein levels were increased in the knee joints of OA patients compared to the non-OA control subjects (Figure 7a). The levels of UBC9 correlated with disease progression, showing an increase in UBC9 in the deep layers of the OA cartilage compared to the superficial layers, which was confirmed by immunohistochemical quantification (Figure 7b).

Of note, the overexpression of *UBE2I* encoding for UBC9 showed that it stabilized PHB1 and promoted its nuclear accumulation in transfected U2OS cells (Figure 8a,c). This mechanism was mediated through the SIM of PHB1, since its deletion in the mutant PHB1_ΔSIM abrogated the effect of UBC9 and significantly reduced PHB1 in the nuclear extracts (Figure 8b).

### 2.4. UBC9 Overexpression in Mice Induces Structural Joint Alterations Characteristic of Primary OA

Because our analysis of human cartilage was limited to end-stage OA, it remains unclear whether elevated UBC9 expression is a primary driver of disease or a secondary consequence. To address this temporal limitation, we utilized a transgenic mouse model overexpressing the Ube2i gene (UBC9) using the CMV early enhancer/chicken β-actin (CAG) promoter, which is widely used for the strong, ubiquitous expression of transgenes across a broad range of tissues, including articular cartilage. This mouse model allowed us to determine whether the sustained elevation of UBC9 is sufficient to initiate or exacerbate OA-like joint degeneration in vivo. Radiographic analysis of knee joints at 35 weeks of age revealed significant macroscopic structural alterations in Tg mice compared to WT controls (Figure 9). Osteophyte formation, a defining feature of OA, was markedly increased in Tg mice, with bony outgrowths observed along the fibula, tibial plateau, femur, and tibial intercondylar eminence (highlighted by yellow squares in Figure 9), indicating widespread osteophyte development. Moreover, a clear reduction in joint space (outlined in red) was evident in Tg mice relative to WT ones, consistent with joint space narrowing (JSN) and cartilage loss. Additional OA-associated features were also apparent in Tg mice, including subchondral sclerosis, reflected by increased radiopacity of the subchondral bone (blue arrows), as well as joint bony erosions (green arrows), indicative of articular bone surface damage. Subchondral cysts and fluid-filled lesions were also detected (purple arrows), further supporting the presence of advanced joint pathology. In contrast, WT mice displayed a largely normal joint architecture, with minimal osteophyte formation, intact joint space, and no detectable signs of subchondral sclerosis, bone erosions, or cyst formation (Figure 9b). These radiographic findings demonstrate that sustained UBC9 overexpression is sufficient to induce macroscopic joint alterations consistent with primary knee OA, supporting a potential causal role for UBC9-driven SUMOylation in disease initiation or progression.

### 2.5. UBC9 Overexpression Impairs Cartilage Integrity and Suppresses Pitx1 Expression in Mouse Knee Joints

To evaluate the effects of elevated UBC9 levels on articular cartilage homeostasis, we performed histological and immunohistochemical analyses on knee joint sections from the UBC9 Tg mice and WT controls. Histological analysis using Safranin-O and Fast Green staining (Figure 10a) revealed a marked reduction in Safranin-O staining intensity in the articular cartilage of Tg mice compared to WT controls. This diminished staining, indicative of reduced proteoglycan content, was accompanied by visible surface irregularities and signs of cartilage fibrillation, suggesting early cartilage matrix degradation. In contrast, WT mice exhibited strong, uniform Safranin-O staining and a smooth articular surface, consistent with preserved cartilage integrity. Immunohistochemical staining for UBC9 (Figure 10b) confirmed robust overexpression of the protein in the articular cartilage of Tg mice. Chondrocytes from the Tg mice displayed intense and widespread dark brown UBC9 staining, while WT cartilage exhibited only minimal UBC9 expression, validating the transgenic model.

Analysis of Pitx1 expression (Figure 10c) revealed a significant downregulation of this chondroprotective transcription factor in Tg mice. Pitx1 staining was substantially weaker and less consistent in Tg cartilage relative to the strong, uniform expression observed in WT controls. Collectively, these results demonstrate that UBC9 overexpression in vivo disrupts cartilage matrix composition and is associated with suppressed *Pitx1* expression. The loss of proteoglycans and reduced levels of this key transcription factor strongly implicate elevated UBC9 activity in promoting cartilage degeneration and osteoarthritis-like changes in the knee joint.

## 3. Discussion

Our previous work established a critical link between Prohibitin 1 (PHB1) and primary knee/hip OA pathogenesis, demonstrating that PHB1 nuclear accumulation in articular chondrocytes coincides with disease onset and directly represses PITX1, a transcription factor essential for cartilage development and homeostasis [19,47]. Although PHB1 is primarily recognized as a mitochondrial protein, forming a complex with PHB2 on the inner mitochondrial membrane to function as a molecular chaperone [48], its mislocalization appears to play a pivotal role in OA. Disruption of the PHB1/PHB2 complex compromises mitochondrial integrity, leading to membrane defects and a reduced cellular lifespan [49], phenotypes observed in OA chondrocytes [50]. Mitochondrial dysfunction has been widely implicated in the disruption of cartilage homeostasis, contributing to metabolic imbalance and accelerating disease progression [51,52]. Our current findings suggest that the nuclear trapping of PHB1 may underlie this dysfunction by depleting its mitochondrial pool, thereby impairing mitochondrial function and linking altered subcellular protein localization to mitochondrial abnormalities in OA. However, the mechanisms underlying PHB1’s aberrant nuclear localization remain unclear.

In this study, we identified SUMOylation, particularly SUMO-1-mediated, UBC9-dependent processes, as a critical axis promoting PHB1 nuclear accumulation. SUMOylation regulates the functions of numerous proteins implicated in human disease [53], including regulators of cartilage homeostasis [54]. It operates via both covalent and non-covalent mechanisms, often altering target proteins’ intracellular localization, stability, or activity [55]. In silico analysis revealed a canonical SUMO-interacting motif (SIM) in PHB1 between amino acids 76 and 80, a region situated between two beta-sheets and not conserved in PHB2, suggesting a selective regulatory mechanism. In vitro SUMOylation assays confirmed that PHB1 is not itself a SUMO substrate; rather, its nuclear accumulation proceeds through non-covalent SIM-dependent interactions with SUMO-1-modified proteins. Importantly, the deletion of the SIM abolished this interaction and almost abrogated nuclear PHB1 levels, establishing the SIM as essential for PHB1’s nuclear retention. Confocal microscopy and immunofluorescence analyses revealed that PHB1 co-localizes with SUMO-1 and accumulates in promyelocytic leukemia nuclear bodies (PML-NBs), stress-responsive subnuclear structures enriched in SUMO-modified proteins [40,41,42,43]. Notably, OA chondrocytes displayed an increased number and size of PML-NBs, sometimes adopting spherical structures indicative of protein recruitment hubs [56]. These findings support a model in which PHB1 is retained in the nucleus via SUMO-1-mediated interactions within PML-NBs. This nuclear trapping may reflect a cellular response to oxidative stress, consistent with the observed formation of PML-NBs in diseased but not healthy tissues [57]. Furthermore, the SUMO-specific protease SENP6, which regulates PML-NB dynamics, was reported to be downregulated in hip OA patients [58]; its depletion is known to enlarge PML-NBs [59]. Taken together, these findings suggest that dysregulated SUMOylation and nuclear architecture remodeling are integral to PHB1 mislocalization in OA.

A central role for the SUMO-conjugating enzyme UBC9 in this process was confirmed by elevated UBC9 protein levels in OA cartilage, particularly in deep cartilage zones correlating with disease severity [60,61]. UBC9 is the sole E2 enzyme required for SUMO conjugation [62], and its overexpression in vitro increased nuclear PHB1 levels, an effect abrogated in SIM-deleted PHB1 constructs. This indicates that UBC9 acts indirectly by enhancing SUMO-1 activity and stabilizing PHB1–SUMO-1 interactions. Notably, the co-expression of UBC9 and SUMO-1 further increased PHB1 nuclear accumulation, while the deletion of PHB1’s SIM abolished this effect, confirming the specificity and requirement of this motif for UBC9-mediated nuclear trapping. To test the biological relevance of this mechanism, we employed a transgenic mouse model expressing, in a systemic manner, the *Ube2i* gene encoding for UBC9. These mice developed pronounced OA-like structural changes by 35 weeks of age, including osteophyte formation, joint space narrowing, subchondral sclerosis, and bone erosions, hallmarks of primary OA. Histological analysis revealed significant proteoglycan loss, matrix degradation, and reduced *Pitx1* expression. These results confirm that increased UBC9 activity and subsequent SUMOylation perturbations can drive structural and molecular changes characteristic of primary knee OA. Our data further support this by showing strong correlations between UBC9 overexpression and global SUMO-1 conjugation levels, with more modest effects on SUMO-2/3 [63]. While a UBC9 knockout model would provide valuable loss-of-function data, complete Ube2i knockout results in early embryonic lethality in mice, precluding global knockout studies for cartilage-specific phenotypes. Furthermore, no cartilage- or chondrocyte-specific Ube2i conditional knockout models have been reported. Despite the current lack of direct gain-of-function studies of UBC9 specifically within the articular cartilage, indirect support comes from other joint disease models. For instance, in a collagen-induced arthritis model of rheumatoid arthritis, UBC9 overexpression was shown to enhance synoviocyte proliferation and migration, contributing to joint inflammation and tissue damage [64,65]. This evidence, combined with our novel UBC9-overexpressing transgenic mouse model, strongly supports the notion that increased UBC9 activity may broadly drive disease-like changes in joint tissues.

Collectively, our findings unveil a previously unrecognized mechanism in OA pathogenesis in which the SUMO-1-dependent, SIM-mediated nuclear retention of PHB1 leads to *PITX1*/*Pitx1* repression and cartilage degeneration in humans and mouse models. Although PHB1 nuclear localization has been observed in other cell types [60], this study is the first to define a SUMO-1-specific, SIM-dependent mechanism for PHB1 nuclear trapping and its functional consequences in primary OA. This mechanism appears specific to PHB1, as PHB2 lacks a conserved SIM and does not show similar localization changes. Our results position UBC9 not merely as a molecular marker, but as a driver of disease progression, offering a novel therapeutic target in OA. In light of mounting evidence that links OA to cellular senescence and stress responses, the role of SUMOylation in modulating protein localization and stability emerges as a critical regulatory axis [53,54,55].

This study offers several novel contributions. First, we delineate a non-covalent, SIM-dependent mechanism for PHB1 nuclear retention in OA chondrocytes. Second, we identify UBC9 as a central upstream driver of this process, validated both in vitro and in vivo. Third, we show that UBC9 overexpression is sufficient to induce OA-like phenotypes in mice, directly linking SUMOylation dysregulation to disease progression. These findings expand the molecular landscape of OA and highlight the SUMOylation machinery as both a biomarker and a potential therapeutic target in early-stage disease. Nonetheless, certain limitations should be acknowledged. While systemic UBC9 overexpression validates a causal role in OA-like pathology, cartilage-specific overexpression would provide more precise insight into cell-type-specific effects. Furthermore, although we propose oxidative stress as a key upstream trigger for SUMO pathway activation, this investigation did not include the direct in vitro exposure of chondrocytes to defined exogenous oxidants. Targeted dose–response and time-course experiments in this context would be valuable to experimentally confirm oxidative stress as a direct driver of PHB1 nuclear trapping. Moreover, the specific SUMOylated nuclear partners mediating PHB1 retention currently remain unidentified. Future proteomic studies are warranted to uncover these key PHB1-bound regulators driving transcriptional repression. Finally, our human data were obtained from end-stage OA cartilage, potentially missing early disease dynamics. Given the challenges in accessing early-stage joint tissue, future studies could utilize surrogate cell types, such as peripheral blood mononuclear cells (PBMCs), which have been proposed as potential sources for detecting OA-related biomarkers like PHB1 [66]. Additionally, longitudinal animal models may help to clarify the temporal sequence of SUMOylation pathway dysregulation, PHB1 mislocalization, and OA onset.

In conclusion, we propose a new model of OA pathogenesis in which UBC9-mediated SUMOylation promotes PHB1 nuclear accumulation, represses *PITX1*, and contributes to cartilage degeneration. This mechanism adds a novel layer of post-translational regulation to OA and suggests new avenues for therapeutic intervention targeting the SUMOylation pathway.

## 4. Materials and Methods

### 4.1. Study Design

Cartilage specimens from tibial plateaus and femoral condyle tissues were obtained from 27 consecutive patients with OA undergoing total knee joint replacement surgery (10 men and 17 women; mean ± SEM age 66 ± 2.8 years) and 4 subjects without OA presenting with trauma (2 men and 2 women, mean ± SEM age 43 ± 13.7 years), as described in Table 1. A certified rheumatologist evaluated all patients with OA according to criteria established by the American College of Rheumatology Diagnostic Subcommittee [67]. Human tissue specimens were collected with the participants’ written consent. The study protocol was approved by the Institutional Research Ethics Boards of CHU Sainte-Justine and Maisonneuve-Rosemont Hospital, Montreal, Quebec, Canada. All participants provided written informed consent. All experiments were performed following relevant guidelines and human ethics regulations.

### 4.2. Human Primary Chondrocyte Cell Preparation

The patient and healthy cartilage samples were dissected, rinsed, and finely minced. The cells were digested with 0.25% trypsin for 1 h at 37 °C. Then, the samples were rinsed with phosphate-buffered saline (PBS) and digested with 2 mg/mL collagenase for 4 to 6 h at 37 °C. The cells were seeded in Falcon culture flasks at a high density (10^6^ cells per 75-cm^2^ flask) and grown to confluence in Dulbecco’s Modified Eagle’s Medium (Gibco™—Thermo Fisher Scientific Inc., Waltham, MA, USA) containing 10% heat-inactivated fetal bovine serum (HyClone—Thermo Fisher Scientific Inc., Waltham, MA, USA) and 1% penicillin and streptomycin at 37 °C in a humidified atmosphere supplemented with 5% CO_2_.

### 4.3. Cell Line

Human osteosarcoma U2OS cells (osteoblast cell line) were cultured in DMEM supplemented with 10% FBS and 1% penicillin and streptomycin. The C28/I2 cells (chondrocyte cell line) were kindly provided by Dr. Mary Goldring (Hospital for Special Surgery, New York, NY, USA) and cultured, as previously described [68]. To generate stable C28/I2 cells, or U2OS cells, overexpressing either PHB1 or the PHB1 construct, retroviral infection was performed using the Phoenix cells generously donated by Dr. Gerardo Ferbeyre (Université de Montréal).

### 4.4. Antibodies and Reagents

Unless specified otherwise, all the reagents were obtained from Sigma-Aldrich Co. LLC.(St-Louis, MI, USA) and BioShop Canada Inc. (Burlington, ON, Canada) and the oligonucleotides were obtained from BioCorp DNA Inc. (Pierrefonds, QC, Canada). For Western blotting, we used antibodies against the following: GAPDH (V-18: sc-20357, dilution [1:5000]) and PHB1 (H-80: sc-28259, dilution [1:500]) (both from Santa Cruz Biotechnology Inc., Dallas, TX, USA); lamin A/C (#2032; dilution [1:2000]), SUMO-1 (A21C7; dilution [1: 2000]), and SUMO-2/3 (18H8; dilution [1:2000]) (all from Cell Signaling Technology Inc., Danvers, MA, USA); FLAG (Sigma M2 F3165; dilution [1:5000]) and RanGap1 (33-0800; dilution [1: 2000]) (both from Zymed Technologies—Thermo Fisher Scientific Inc., Waltham, MA, USA); and GST (M1; dilution [1: 2000]) (from EMD Millipore, Burlington, MA, USA). For immunohistochemical analysis, we used anti-UBC9 (N-15; dilution [1:100]) (from Santa Cruz Biotechnology Inc., Dallas, TX, USA). For immunofluorescence analysis, we used antibodies against PHB1 (Ab-1; dilution [1:50]) (from Lab Vision™—Thermo Fisher Scientific Inc., Waltham, MA, USA); TOM-20 (FL-145: sc-11415; dilution [1:50]) and PML (PG-M3; dilution [1:25]) (both from Santa Cruz Biotechnology Inc., Dallas, TX, USA); and SUMO-1 (A21C7; dilution [1:200]) and SUMO-2/3 (18H8; dilution [1:100]) (both from Cell Signaling Technology, Inc., Danvers, MA, USA). For immunoprecipitation, the following antibodies were used: anti-PHB1 (N-20; dilution [1:100]) (from Santa Cruz Biotechnology, Inc., Dallas, TX, USA) and anti-c-Myc (MAB8865; dilution [1:200]) (from EMD Millipore, Burlington, MA, USA).

### 4.5. Vector Constructs

The various PHB1 mutants described below were constructed from wild-type PHB1 cDNA (OriGene Technologies, Inc.,Rockville, MD, USA), which was amplified and cloned in-frame in a plasmid (pLPC) vector containing thrice-repeated FLAG epitopes [69], generously donated by Dr. GerardoFerbeyre. The 3xFlag tag was inserted into the N-terminal of wild-type PHB1 and the mutants. The pCDNA3-Myc-SUMO1, pCDNA-Myc-SUMO-3, and pCDNA-ha-SUMO2 plasmids were kindly provided by the laboratory of Dr. Christopher K. Knell (University of California, San Diego, CA, USA). The pcDNA3.1-Gal4-UBC9 vector was also kindly provided by Dr. Muriel Aubry (Université de Montréal, Montreal, QC, Canada). The PCR primer sequences used were as follows: 3xFlag tagged PHB1—5′-GCGAATTCTGCTGCCAAAGTGTTTGAGTCCATTGGC (forward) and 5′-GCCTCGAGTCACTGGGGCAGCTGAGGA (reverse); PHB1-∆SIM (a putative sumo binding module (residues 76 to 79))—5′-CCACGTAATACTGGTAGCAAAGATTTACAGAATGTC-3′ (forward) and 5′-GCTACCAGTATTACGTGGTCGAGAACGGCAGTCA-3′ (reverse); and PHB1-ΔNLS—a nuclear export signal was replaced by a nuclear localization signal using the following primers: 5′-GCCTCGAGTCAGCCCACTTTGCGCTTTTTCTTGGGG-3′ (forward) and 5′-TTCCGAGAGCGTGAGAGCTGGTA-3′ (reverse).

### 4.6. Nuclear and Cytoplasmic Protein Isolation

Three 100 mm confluent plates (per condition) of primary chondrocyte cells or U2OS cells were washed twice in ice-cold PBS, scraped, and centrifuged for 5 min at 1000× *g*. The cell pellets were resuspended in 300 µL of hypotonic lysis buffer (10 mM Hepes pH 7.9; 1.5 mM MgCl_2_; 10 mM KCl; 1% NP-40; 0.5 mM DTT) supplemented with 1× protease inhibitors and 25 mM of N-Ethylmaleimide (NEM). They were then incubated on ice for 25 min by vortexing every 3 to 4 min. The lysates were centrifuged for 5 min at 4 °C at 1200× *g* to collect the nuclear extract. The supernatants containing cytoplasmic proteins were centrifuged once more to remove nuclear protein contaminants. The nuclear extract was then resuspended in 8 mL of nuclear lysis buffer (50 mM Tris-HCl pH 7.6; 2 mM EDTA; 2 mM EGTA; 1 mM DTT; 1X protease inhibitors and 25 mM of NEM containing 0.1% Triton X-100) and overlaid on a 2 mL cushion of sucrose (nuclear lysis buffer containing 30% *w/v* sucrose) in 15 mL tubes. The samples were centrifuged at 4 °C for 50 min at 3500× *g*. The buffer was discarded, and the purified nuclear extract was resuspended in 50–100 µL of 4× Laemmli buffer and boiled for 5 min. The protein concentration was measured using the Bradford method (Bio-Rad, Hercules, CA, USA), and 50 µg each of the cytoplasmic and nuclear proteins were separated by SDS-PAGE and analyzed by Western blotting.

### 4.7. Co-Immunoprecipitation Assays

The U2OS cells were co-transfected with pCMV4-Myc-Sumo1 in the presence of pLPC-3xFlag-PHB1 or pLPC-3xFlag-PHB1-∆SIM (15 µg of total DNA) by the calcium phosphate method. Following 48 h of transfection, the cells were lysed directly in a lysis buffer (20 mM Tris pH 7.8, 0.15 M NaCl, 1 mM EDTA, 1% Triton X-100, 2.5 mM Sodium pyrophosphate, 1 mM β-glycerolphosphate), 1x protease inhibitor mixture, and 25 mM NEM. Following 60 min of incubation at 4 °C with gentle shaking, the lysates were collected and centrifuged at 15,000× *g* for 15 min at 4 °C. The supernatants were further subjected to overnight immunoprecipitation using 1–2 mg of total proteins at 4 °C and were collected using the proteins A/G sepharose (GE Healthcare Bio-Sciences, Malborough, MA USA) at 4 °C for one hour. The samples were washed three times with 1× PBS and one time with water. The samples were eluted in 70 µL of 3× Laemmli buffer, boiled for 5 min, and 35 µL was used for Western blot analysis.

### 4.8. GST-PHB1 and GST-RanGap1 Purification

The GST-PHB1 and GST-RanGap1 proteins were expressed in bacteria and isolated as follows: the bacteria were collected by centrifugation and resuspended in lysis buffer (10 mM Tris-HCl pH 8, 1 mM EDTA, 100 mM NaCl, 5 mM DTT, 1 mM PMSF and 1× protease inhibitor mixture). In total, 1 mg/mL of lysozyme was added, and the samples were incubated on ice for 30–45 min; then, 1.5% sarcosyl was added, and the samples were sonicated. After centrifugation of the cell lysates, the supernatants were transferred into new tubes. The GST fusion proteins were isolated using glutathione-coupled Sepharose 4B beads according to the manufacturer’s instructions (GE Healthcare Bio-Sciences, Malborough, MA, USA). SDS-PAGE and Coomassie blue staining analyzed the GST fusion proteins.

### 4.9. In Vitro SUMOylation Assay

Purified GST, GST-PHB1, and GST-RanGap1 proteins were used as substrates for the in vitro SUMOylation assay using SUMO1 protein. GST and GST-RanGap1 were used as negative and positive controls, respectively. Each reaction was carried out in a total volume of 20 µL in a reaction buffer containing 20 mM HEPES pH 7.5 and 5 mM MgCl_2_, with the presence of 7.5 µg/mL of the E1 enzyme, 50 µg/mL of E2, 50 µg/mL of SUMO1, and 20 mM ATP, and was incubated for 1 h at 37 °C. All the reagents were obtained commercially (LAE Biotech International, Taipei, Taiwan) and were used according to the manufacturer’s instructions. A 5 µL sample was separated by SDS-PAGE for each reaction and analyzed by Western blotting.

### 4.10. Immunohistochemistry Analysis

The cartilage sections from human tissue were embedded in paraffin, and 5 μm sections were cut using Microtome (Leica rm2145, Wetzlar, Germany). For immunohistochemical analysis, the sections were rehydrated and heated at 65 °C for 15 min in 0.01 M citrate buffer, rinsed in PBS, and put into 0.3% Triton-PBS solution for 30 min for antigen retrieval. The tissue sections were soaked in a solution containing 2% hydrogen peroxide in methanol for 30 min to inactivate endogenous peroxidase. The sections were rewashed, blocked for 15 min in PBS with 1% bovine serum albumin, and incubated overnight at 4 °C with primary antibody against UBC9 for human cartilage sections. Following PBS washes, the tissue sections were incubated with a secondary biotinylated antibody for 45 min. The staining was revealed using the avidin–biotin complex method (Vectastain ABC kit—Vector Laboratories, Newark, CA, USA) and the 3,3′-diaminobenzidine (DAB) detection system (Dako North America Inc.—Agilent Technologies, Santa Clara, CA, USA). The sections were counterstained with Harris Modified Hematoxylin (Thermo Fisher Scientific Inc., Waltham, MA, USA).

### 4.11. Immunofluorescence Analysis

Human chondrocytes from OA and non-OA (control) patients were plated on 8-well glass culture slides (BD Biosciences, Franklin Lakes, NJ, USA) with 10^4^ cells per well. After 24 to 48 h of incubation, the cells were washed with PBS, fixed with 3.7% paraformaldehyde solution, and permeabilized with 0.1% Triton X-100 in PBS for 10 min. The cells were then blocked in 1% bovine serum albumin in PBS for 30 min at room temperature and incubated with primary antibodies for 2 h at 37 °C. Secondary antibodies conjugated to Alexa fluor dyes (Invitrogen™—Thermo Fisher Scientific Inc., Waltham, MA, USA) were diluted in bovine serum albumin in PBS and applied for 1 h at 37 °C. After a PBS wash, the cells were mounted using the ProLong Gold antifade reagent and the DAPI fluorescent stain (Invitrogen™—Thermo Fisher Scientific Inc., Waltham, MA, USA). The slides were analyzed using the Zeiss LSM 510 Meta microscope (Germany; Zeiss Canada, Toronto, ONE, Canada) at 630× magnification. The images were then analyzed using the software Zeiss LSM image browser, version 4.

### 4.12. In Silico Analysis

The NCBI reference sequence of PHB1 is [NP_002625.1]. We used the GPS-SUMO website (https://sumo.biocuckoo.cn, accessed on 18 May 2009) for in silico analyses [70,71]. We selected [all] parameters for the threshold for both SUMOylation and SUMO interaction. For the scheme figure, we chose [visualize] as the parameter.

### 4.13. Animal Model

To further investigate the role of UBC9 in OA pathogenesis in vivo, we utilized well-characterized UBC9-overexpressing transgenic mouse lines (Tg-UBC9 N2 and H3) originally generated at the National Institute of Mental Health (NIMH) [63], and frozen sperm samples were graciously provided by Dr. John M. Hallenbeck (NIMH). Briefly, the mouse Ub2ei coding sequence was inserted between the Kpn1 and Xho1 sites of pCCALL2-anton DlacZ. The Ub2ei gene was located under control of the CMV early enhancer/chicken β-actin (CAG) promoter and followed by the rabbit β-globin poly adenylation sequence. These transgenic strains were rederived from cryopreserved sperm (Transgenesis Unit at the Research Center of Université de Montreal Hospital/CRCHUM). Once established, these transgenic mouse strains were maintained at the Azrieli Research Center animal facility at CHU Sainte-Justine. The C57/BL6 mice were purchased from Charles River Laboratories (Senneville, QC, Canada) and also housed at the Azrieli Research Center animal facility at CHU Sainte-Justine. Mice were housed in groups of 3–5 per ventilated cage under standard conditions. No intervention or specific treatment was administered; the mice were allowed to age naturally until approximately 35 weeks of age to mimic the OA phenotype typically observed in elderly patients. No specific inclusion or exclusion criteria were applied for mouse selection, and potential confounders were not controlled. Humane endpoints, including significant weight loss and/or inability to eat or drink, were monitored daily (no exclusions were made). A total of 6 mice per sex and per strain were used to reach statistical power. All experimental procedures were reviewed and approved by our Animal Institutional Review Board [Comité des Bonnes Pratiques Animals et Recherche (CIBPAR)], ensuring adherence to standard regulations for the care and use of laboratory animals.

These transgenic mice were utilized to investigate the impact of increased UBC9 expression on the development of OA-related phenotypes (Table 2). The UBC9-overexpressing group (n = 12) had an average age of 38 ± 1.1 weeks and consisted of an equal number of males and females (6:6). The average weight for this group was 30 ± 2.1 g. The control group (C57BL/6/wild type, n = 12) had an average age of 33 ± 2.4 weeks and also consisted of an equal number of males and females (6:6). The average weight for the control group was 33 ± 2.3 g. Statistical analysis revealed no significant differences in either age or weight between the UBC9-overexpressing and control groups. An average age of 35 weeks was used for all mice combined.

### 4.14. Radiographic Evaluation of Knee Joints in the Mice

To assess macroscopic structural changes indicative of OA progression in the knee joints of the UBC9-overexpressing and control mice, a radiographic evaluation of the 12 UBC9 mice and 12 wild type mice was performed once at the experimental endpoint. The animals were euthanized; radiographs of the knee joints were obtained using the Faxitron X-ray system at standardized settings to ensure a consistent image quality. The severity of OA-related changes was assessed based on parameters evaluated, typically including osteophyte formation (presence and size at joint margins), subchondral sclerosis (increased bone density beneath cartilage), and joint space narrowing (reduced distance between bones).

### 4.15. Immunohistochemistry of Articular Cartilage

The hind limbs of 6 UBC9 and 6 wild type mice were fixed in 4% paraformaldehyde (PFA) in 0.1M phosphate buffer (pH 7.4) for 48 h. Following fixation, the limbs were washed three times with phosphate-buffered saline (PBS) and decalcified in 10% ethylenediaminetetraacetic acid (EDTA) (pH 8.0) for two weeks with gentle agitation, ensuring complete calcium removal. Decalcified hind limbs were then embedded in paraffin, sectioned at a 5 µm thickness using a microtome, and mounted onto Superfrost-plus slides. The slides were stored at room temperature until labeling. Prior to immunostaining, the tissue sections were deparaffinized in xylene and rehydrated through a series of decreasing ethanol concentrations (e.g., 100%, 95%, and 70%). Antigen retrieval was performed by immersing the slides in 10 mM citrate buffer (pH 6.0) and heating them at 65 °C for 20 min. To further enhance antibody penetration, the sections were treated with trypsin from porcine pancreas (1 mg/mL; Sigma-Aldrich, T0303) for 15 min at 37 °C, followed by three washes with PBS. Non-specific antibody binding was blocked by incubating the sections in 5% bovine serum albumin (BSA) in PBS containing 0.1% Triton X-100 for 1 h at room temperature. The slides were then incubated overnight at 4 °C with primary antibodies diluted in antibody dilution buffer (Reagent G, Elabscience, Houston, TX, USA, E-IR-R220). The primary antibodies used were a monoclonal anti-UBC9 antibody (1:100 dilution, Thermo Fisher, 60201-1-IG) and a polyclonal rabbit anti-PITX1 antibody (1:100 dilution, Thermo Fisher, 10873-1-AP). Following primary antibody incubation, the sections were washed three times with PBS and endogenous peroxidase activity was blocked by incubation with peroxidase blocking buffer (Reagent B, Elabscience, E-IR-R220) for 15 min. After three further washes with PBS, the sections were incubated with a Polyperoxidase anti-rabbit/mouse IgG secondary antibody (Reagent C, Elabscience, E-IR-R220) for 1 h at room temperature. The signal was visualized using a DAB working solution prepared by diluting a highly sensitive DAB concentrate (Reagent D, Elabscience) 1:10 in a highly sensitive DAB substrate (Reagent E, Elabscience) for 3 min, with careful monitoring of color development. The reaction was stopped by washing the slides three times with PBS. Sections were then counterstained with Hematoxylin, followed by a brief immersion in alkaline water (1% lithium carbonate) to enhance blue staining, and thoroughly rinsed with deionized (DI) water. Finally, the slides were dehydrated through a graded ethanol series, cleared in xylene, and mounted with Microkit mounting media (Mecalab). Images were acquired using a digital slide scanner (Zeiss Axioscan) and analyzed using ZEISS Zen 3.1 Blue Edition software.

### 4.16. Safranin-O Staining of Articular Cartilage

To evaluate the structural integrity and proteoglycan content of the articular cartilage in the animal model, knee joint sections were stained with Safranin-O and Fast Green. Following deparaffinization and rehydration (as described above), the sections were first stained with Weigert’s hematoxylin to visualize cell nuclei. Subsequently, non-cartilaginous tissues were counterstained with a 0.04% Fast Green solution. Finally, the cartilage matrix was stained with a 0.1% Safranin-O solution for a precisely controlled duration of 3 min. Safranin-O, a cationic dye, binds specifically to the negatively charged glycosaminoglycans (GAGs) within the proteoglycans of the cartilage matrix, resulting in an orange to red staining. The intensity of Safranin-O staining is directly indicative of the proteoglycan concentration. After staining, the sections were dehydrated through a series of increasing ethanol concentrations, cleared in xylene, and permanently mounted. Images were acquired using a Zeiss Axio Scan.Z.1 slide scanner and analyzed with ZEISS Zen 3.1 Blue Edition software. Histological assessment of the Safranin-O/Fast Green-stained sections was performed to evaluate the extent of cartilage degradation, characterized by parameters such as loss of Safranin-O staining intensity, surface fibrillation, and chondrocyte depletion.

### 4.17. Statistical Analysis

Data are presented as mean ± standard error of the mean (SEM). Statistical comparisons between the two groups were performed using unpaired Student’s *t*-tests. For the analysis of radiographic scores, histological grading, and immunohistochemical quantification, observers were blinded to the group allocation of the samples.

## Figures and Tables

**Figure 1 ijms-26-06281-f001:**
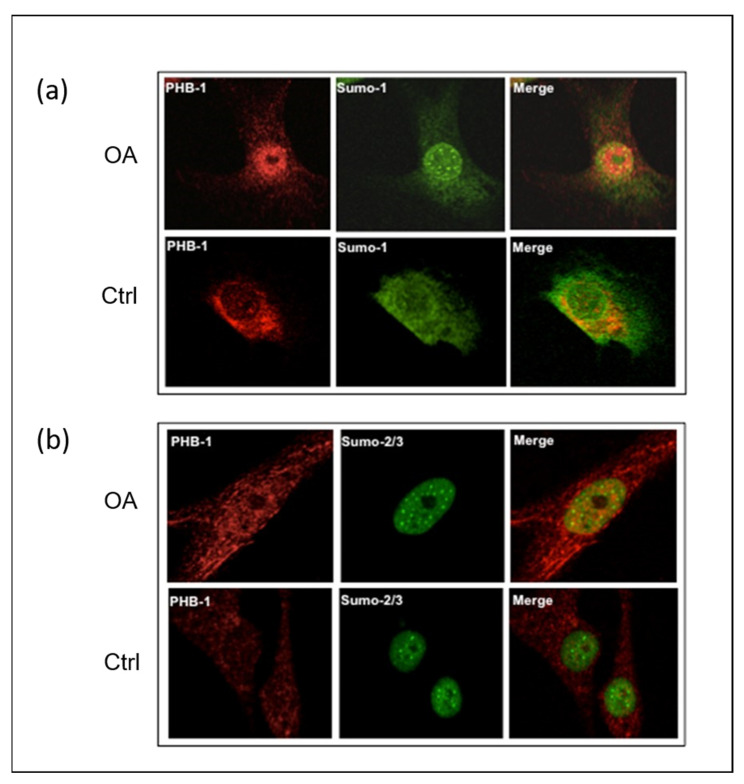
SUMO-1 proteins accumulate in the nuclei of OA articular chondrocytes and co-localize with PHB1. Representative double immunofluorescence staining against PHB1 (red), SUMO-1 (green), and SUMO-2/3 (green) was carried out on articular chondrocytes of one OA patient and one healthy subject. In OA patients, PHB1 shows a distinct accumulation in the nucleus, in contrast to control cells where it is primarily cytoplasmic. Furthermore, the SUMO proteins accumulate in nuclear bodies (NBs) in OA, while control patient cells show little or no accumulation of SUMOs in the nuclear bodies. (**a**) In OA chondrocytes, PHB1 (red) is co-localized with SUMO-1 (green), appearing as a strong yellow signal in a merged picture (upper panel). In control cells, PHB1 is largely cytoplasmic with minimal nuclear presence and SUMO-1 nuclear bodies are less prominent (lower panel). (**b**) In OA chondrocytes, a nuclear accumulation of PHB1 (red) is shown, and is co-localized with SUMO-2/3 (green) (upper panel). In control cells, PHB1 remains predominantly cytoplasmic, and SUMO-2/3 nuclear bodies are less defined (lower panel).

**Figure 2 ijms-26-06281-f002:**
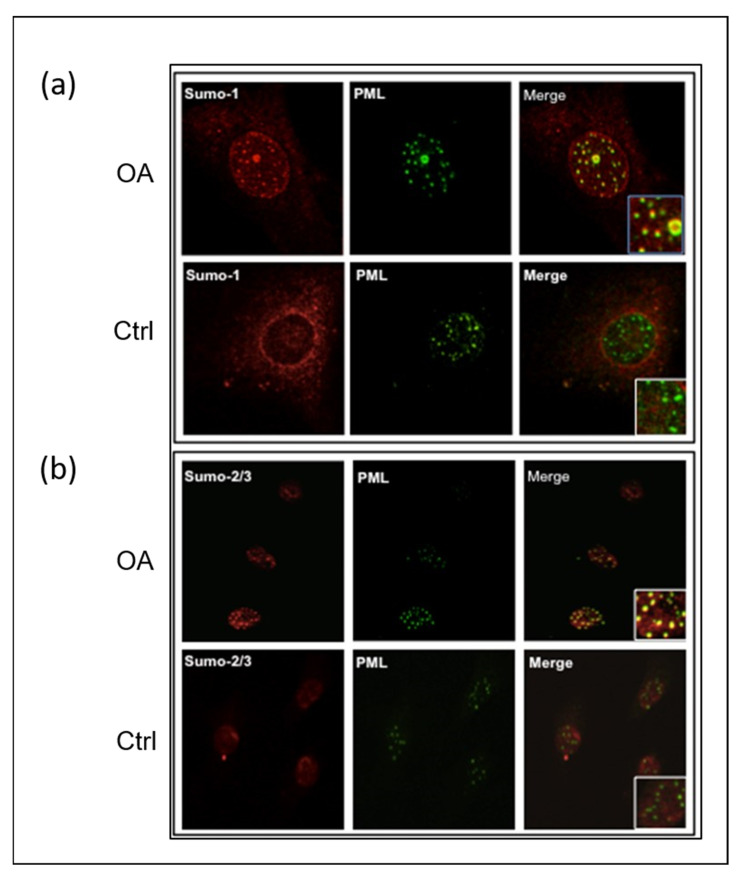
SUMO proteins accumulate in PML nuclear bodies in OA articular chondrocytes. Representative double immunofluorescence staining on human articular chondrocytes of OA patients and control subjects was performed with antibodies against PML (green), SUMO-1 (red), and SUMO-2/3 (red). (**a**) SUMO-1 is co-localized in the PML nuclear bodies of OA chondrocytes. The accumulation of SUMO-1 is in the nucleus, including the PML nuclear bodies. (**b**) SUMO-2/3 is co-localized in the PML nuclear bodies of OA chondrocytes. In OA chondrocytes, nuclear accumulation of SUMO-2/3 proteins is mainly localized in the PML nuclear bodies.

**Figure 3 ijms-26-06281-f003:**
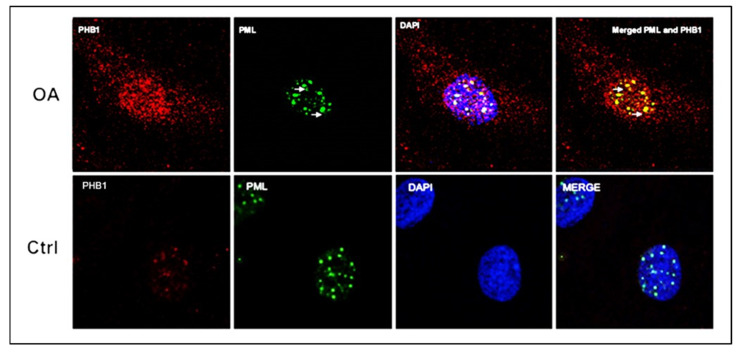
PML and PHB1 co-localize in OA patients’ nuclei of human articular chondrocytes. Representative double immunofluorescence staining on human articular chondrocytes of OA patients and control subjects was performed with antibodies against PML (green) and PHB-1 (red). The nucleus is stained blue with DAPI. The upper panel (OA) shows that PHB1 accumulates mainly in the nuclei of OA chondrocytes like PML, which colocalizes with the PML nuclear bodies. The lower panel (Ctrl) shows that PHB1 is not detected in the nuclei of control human articular chondrocytes.

**Figure 4 ijms-26-06281-f004:**
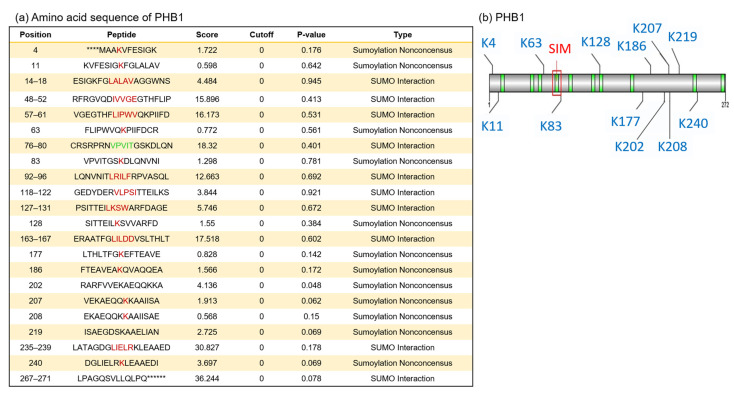
In silico analysis of primary amino acid sequences of human PHB1 protein reveals putative SUMOylation sites and SUMO-binding motifs. We used GPS-SUMO (https://sumo.biocuckoo.cn/, accessed on 18 May 2009) for in silico analyses. We selected [all] for SUMOylation and SUMO interaction (SIM) for the threshold. (**a**) The location of all probabilities of putative SUMOylation associated with each site in PHB1 (red) and for the selected SIM (green). The asterisks (*) indicate amino acids that precede or follow the PHB1 sequence but are not shown. (**b**) The scheme of SUMOylation sites (blue), SUMO interaction (SIMs) (green), and the selected SUMO-binding site for SIM in PHB1 (red square).

**Figure 5 ijms-26-06281-f005:**
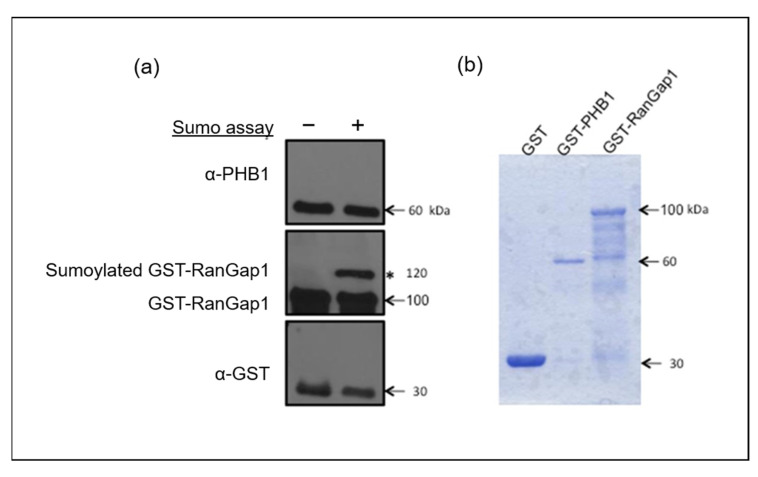
SUMO1 cannot SUMOylate PHB1 in vitro. An in vitro SUMOylation assay in the presence of SUMO-1, E1 and E2 enzymes, ATP, and purified GST-PHB-1 protein indicated that PHB1 could not be SUMOylated in vitro. GST and GST-RanGap1 proteins were used as negative and positive controls, respectively. (**a**) Four times less GST protein was used for the test than the fusion proteins. The in vitro SUMOylation assay products were analyzed by immunoblots against PHB1 and RanGap1. The asterisk (*) represents the SUMOylated GST-RanGap1. (**b**) The purified GST and GST fusion proteins were analyzed using SDS-PAGE followed by Coomassie blue staining.

**Figure 6 ijms-26-06281-f006:**
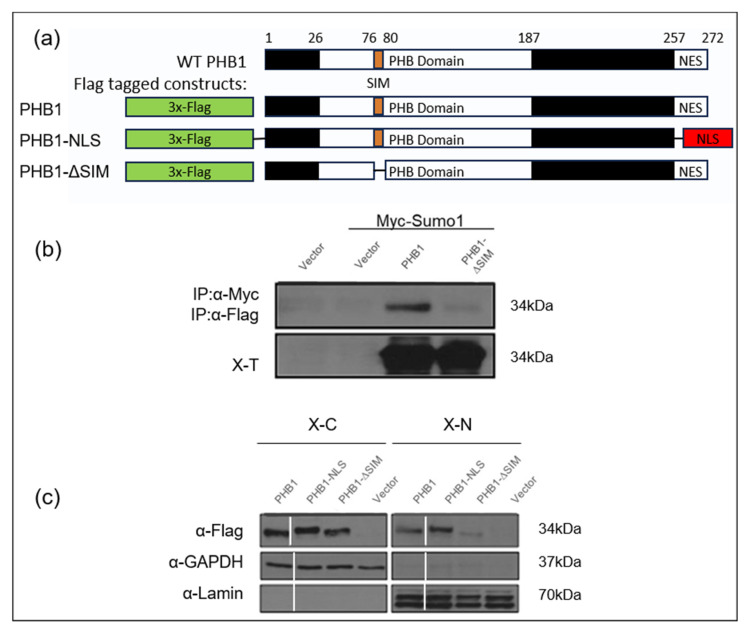
PHB1 can bind SUMO1 proteins via a SIM (SUMO-interacting module), which is crucial for its nuclear localization. (**a**) Diagram represents wild-type PHB1 protein structure and various PHB1 constructs: wild-type PHB1 (WT PHB1), a mutant where the nuclear signal of export was deleted (PHB1-∆NES) or was replaced by a nuclear localization signal (PHB1-NLS), and a mutant where a putative SUMO-interacting motif (SIM) was deleted (PHB1-∆SIM). All the constructs have a triple Flag-tag at the N-terminal. (**b**) Co-immunoprecipitation assays with anti-c-Myc antibodies demonstrate that PHB1 interacts with Myc-tagged SUMO1 through the SIM (upper panel). The lower panel indicates the level of Myc-tagged SUMO1 protein in total cell extracts (X-T). (**c**) The nuclear accumulation of PHB1 is dependent on its SIM. C28/I2, a human chondrocyte cell line, were infected with either flag-tagged wild type (PHB1), (PHB1_NLS), or (PHB1_∆SIM) constructs or empty vector, to produce stable cell lines. The nuclear extract (X-N) and the cytoplasmic extract (X-C) proteins were isolated and analyzed by Western immunoblotting to detect the subcellular presence of flag-tagged PHB1. Anti-GAPDH was used as a cytoplasmic loading control, and anti-Lamin was used as a nuclear loading control. Note the significantly reduced nuclear presence of PHB1-ΔSIM compared to WT PHB1 and PHB1-NLS.

**Figure 7 ijms-26-06281-f007:**
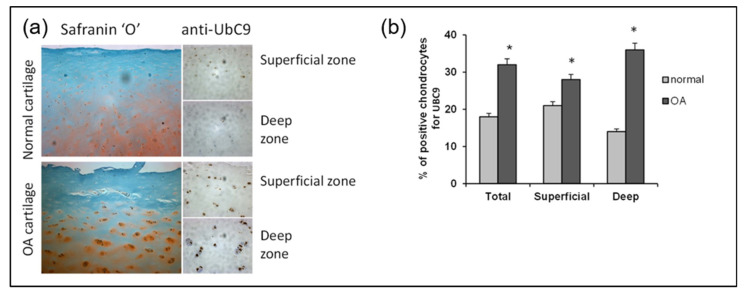
UBC9 expression is increased in the OA cartilage of the knee joint and correlates with the disease severity. (**a**) Representative immunohistological sections showing UBC9 in the human articular cartilage of control and OA subjects. The left panels show Safranin-O staining (red), which represents the proteoglycan content, which decreases with the severity of OA. The right panels represent IHC staining in superficial and deep zones of normal and OA human articular cartilage, performed with anti-UBC9 antibody (brown signal), where the staining intensity correlates with disease progression. (**b**) Immunohistochemical quantification of UBC9 in the superficial and deep zones of human cartilage in normal (n = 3) and OA (n = 12). Data are shown as mean ± SEM per condition. * *p* < 0.05. The scale bar is 100 µm.

**Figure 8 ijms-26-06281-f008:**
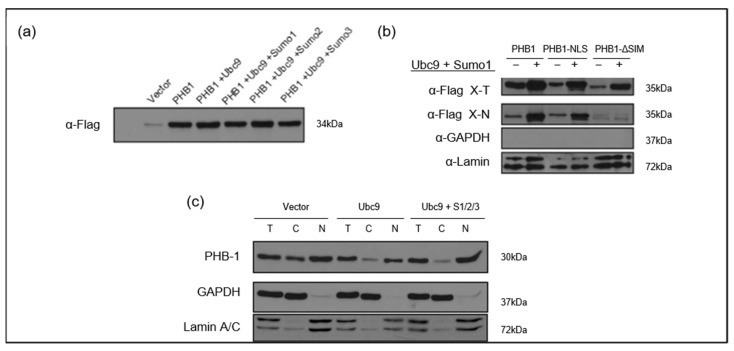
The UBC9-mediated SUMO pathway stabilizes PHB1 and promotes its nuclear accumulation in U2OS cells. (**a**) Co-expression of UBC9 and SUMO isoforms enhances PHB1 protein levels. U2OS cells were transfected with the pLPC-3xFlag-PHB1 alone or co-transfected with different components of the SUMOylation pathway (UBC9; UBC9 + Sumo1; UBC9 + Sumo2; UBC9 + Sumo3). Total cell lysates were analyzed by Western blotting using an anti-Flag antibody to detect Flag-PHB1 protein levels. (**b**) The nuclear accumulation of PHB1 is dependent on its SIM in the presence of UBC9 and SUMO-1. U2OS cells were transfected with Flag-tagged PHB1, PHB1-NLS, or PHB1-ΔSIM constructs, in the presence or absence of co-transfected Myc-SUMO-1 and UBC9. The nuclear proteins (=X-N), as well as total proteins (X-T), were isolated from cells transfected with pLPC-3xFlag-PHB1, PHB1-NLS, or PHB1-∆SIM in the presence or absence of myc-SUMO1 and UBC9. Anti-GAPDH was used as a cytoplasmic loading control, and anti-Lamin was used as a nuclear loading control, demonstrating successful cell fractionation. (**c**) U2OS cells were transfected with UBC9 alone or co-transfected with pCMV4-myc-SUMO1, HA-SUMO2, and myc-SUMO3 or with the empty vector. Total (T), cytoplasmic (C), and nuclear (N) protein extracts were isolated. Western blot analysis using an anti-PHB1 antibody reveals changes in endogenous PHB1 subcellular localization. Anti-GAPDH and Anti-Lamin A/C were used as loading controls for cytoplasmic and nuclear fractions, respectively.

**Figure 9 ijms-26-06281-f009:**
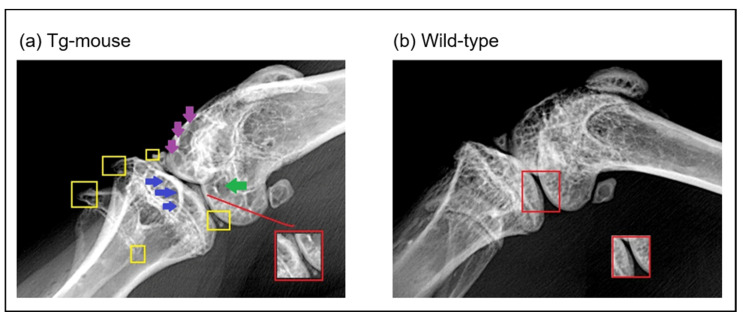
X-ray radiographic analysis of knee joints at 35 weeks in wild-type and UBC9-overexpressing transgenic mice. Representative radiographs of knee joints from (**a**) UBC9-overexpressing transgenic (Tg) and (**b**) wild-type (WT) mice at 35 weeks of age, obtained using a soft X-ray apparatus (Faxitron), illustrating macroscopic structural changes associated with osteoarthritis. The Tg mouse knee joints exhibit significant alterations compared to WT controls, including increased osteophyte formation (yellow squares) along the fibula, tibial plateau, femur, and tibial intercondylar eminence; increased joint space narrowing (JSN) (red square); obvious signs of subchondral sclerosis (blue arrows); joint bony erosions (green arrows); and subchondral cysts (violet arrows).

**Figure 10 ijms-26-06281-f010:**
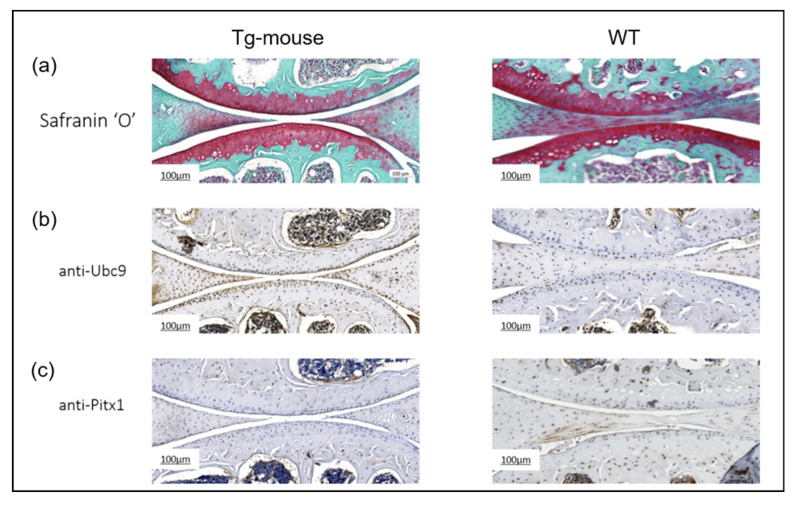
Histological and immunohistochemical analysis of knee joints from wild-type and UBC9-overexpressing transgenic mice. (**a**) Safranin-O and Fast Green Staining: Representative histological images of whole knee joint sections from UBC9 transgenic and wild-type mice at 35 weeks of age stained with Safranin-O (red) to visualize proteoglycan content and Fast Green (counterstain). Note the reduced Safranin-O staining intensity in the articular cartilage of the Tg mouse, indicative of proteoglycan loss and cartilage degradation, compared to the strong and uniform staining in the WT mouse. Scale bar = 100 μm. (**b**) Immunohistochemistry for UBC9: Representative images showing immunohistochemical staining for UBC9 protein (brown signal) in knee joint sections from Tg and WT mice. The Tg mouse exhibits markedly stronger and more widespread UBC9 staining within chondrocytes compared to the minimal staining observed in the WT control. Scale bar = 100 μm. (**c**) Immunohistochemistry for Pitx1: Representative images showing immunohistochemical staining for Pitx1 protein (brown signal) in knee joint sections from Tg and WT mice. A substantial downregulation of Pitx1 staining intensity is evident in the chondrocytes of the Tg mouse compared to the more consistent and intense Pitx1 staining in the WT cartilage. Scale bar = 100 μm. All results are representative of experiments conducted with WT (n = 12) and Tg (n = 12) mice.

**Table 1 ijms-26-06281-t001:** Demographic characteristics of the study participants.

	OA	Matched Non-OA Controls (Trauma Cases)
N	27	4
Women/men	17/10	2/2
Age	66 ± 2.8	43 ± 13.7

All data are presented as mean and ± standard error of the mean (SEM). A T-test was performed to determine the significance level between the two groups. Abbreviation Osteoarthritis (OA).

**Table 2 ijms-26-06281-t002:** Comparative table of animal characteristics.

Characteristic	UBC9 Group (n = 12)	C57BL/6/Wild Type Group (n = 12)
Average Age (Weeks)	38 ± 1.1	33 ± 2.4
Sex Ratio (M:F)	6:6	6:6
Average Weight (g)	30 ± 2.1	33 ± 2.3

Data are presented as the mean ± standard error of the mean (SEM).

## Data Availability

The original data presented in this study are included in the article. Further inquiries can be directed to the corresponding author.

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
