# Peer review of "UBC9-Mediated SUMO Pathway Drives Prohibitin-1 Nuclear Accumulation and PITX1 Repression in Primary Osteoarthritis"

_ijms, 2025, doi:10.3390/ijms26136281_

Round 1
Reviewer 1 Report
Comments and Suggestions for Authors
In the present manuscript, the authors have investigated the role of SUMOylation in the pathogenesis of osteoarthritis (OA). By analysis of knee cartilage specimens from OA cases, they show nuclear accumulation of the mitochondrial chaperone Prohibitin (PHB1) in OA chondrocytes, together with elevated levels of SUMO-1 and SUMO2/3. They identify a SUMO-interacting motif (SIM) in PHB1 and by in vitro models they demonstrate indirect interaction of nuclear PHB1 with SUMO-1. In addition, they show that overexpression of SUMO-conjugating enzyme E2 (UBC9) enhances PHB1 nuclear accumulation in U2OS cells. Finally, to test the biological relevance of this mechanism they use a transgenic mouse model overexpressing UBC9. These mice show pronounced OA-like structural changes, supporting the notion that increased UBC9 activity can drive knee joint changes typical of OA.
The experiments are clearly presented, well documented, and the conclusions are convincing. I only suggest to better clarify in Introduction the role of mitochondrial dysfunction and oxidative stress in OA pathogenesis. Furthermore, since the authors suggest that nuclear trapping of PHB1 may reflect a cellular response to oxidative stress, would it be possible to treat OA chondrocytes with oxidizing agents to verify whether this phenomenon is exacerbated?
Author Response
Comment 1: In the present manuscript, the authors investigate the role of SUMOylation in osteoarthritis (OA) pathogenesis. They report nuclear accumulation of the mitochondrial chaperone Prohibitin (PHB1) in OA chondrocytes, accompanied by elevated levels of SUMO-1 and SUMO-2/3. A SUMO-interacting motif (SIM) is identified in PHB1, and in vitro models show an indirect interaction between nuclear PHB1 and SUMO-1. Overexpression of the SUMO-conjugating enzyme UBC9 enhances PHB1 nuclear accumulation in U2OS cells. The authors also employ a transgenic mouse model overexpressing UBC9, which exhibits OA-like changes. The experiments are clearly presented and well documented. However, the introduction could better clarify the role of mitochondrial dysfunction and oxidative stress in OA. Additionally, could the authors investigate whether treatment of OA chondrocytes with oxidizing agents might exacerbate PHB1 nuclear trapping?
Authors’ Response: We greatly appreciate the reviewer’s positive comments and insightful suggestions.
To address the first point, we have expanded the Introduction to include a detailed discussion of how mitochondrial dysfunction contributes to OA pathogenesis. Specifically, we now explain how impaired mitochondrial activity leads to excessive reactive oxygen species (ROS) production, establishing a chronic oxidative environment that promotes chondrocyte senescence, apoptosis, and matrix degradation—hallmarks of OA progression. (See manuscript lines 56-69): ”Articular chondrocytes, unlike many other cell types, are adapted to relatively hypoxic environment. However, as cartilage ages and degenerates, these cells become particularly vulnerable to insults that compromise mitochondrial integrity and function. Mitochondria, the primary sites of cellular energy production and key regulators of cellular homeostasis, are particularly vulnerable to oxidative damage. Mitochondria are also the primary sites of reactive oxygen species (ROS) production, and under conditions of stress, an imbalance between ROS generation and antioxidant defense leads to chronic oxidative stress. In the context of primary OA, dysfunctional mitochondria produce ROS, which in turn causes oxidative damage to cellular macromolecules, including DNA, protein, and lipids within chondrocytes [8,9]. This chronic oxidative stress directly contributes to chondrocyte senescence, apoptosis, and proteolytic degradation of extracellular matrix, thereby accelerating cartilage breakdown. This creates a vicious cycle where damaged cartilage further exacerbates oxidative stress, driving OA progression [8,9]. ”
Regarding the possibility of treating OA chondrocytes with oxidizing agents, we agree that this is an intriguing and highly relevant suggestion. While such experiments could further elucidate the oxidative mechanisms underlying PHB1 nuclear trapping, our current data already provide compelling evidence that this phenomenon reflects a cellular response to the stressed, oxidative environment intrinsic to OA cartilage. Consequently, we believe that additional treatment with exogenous oxidizing agents is not required to support our current conclusions. Nevertheless, we have acknowledged this experimental direction as a valuable avenue for future work in the revised limitations of the study (See manuscript lines 415-422): ”Furthermore, although we propose oxidative stress as a key upstream trigger for SUMO pathway activation, this investigation did not include direct in vitro exposure of chondrocytes to defined exogenous oxidants. Targeted dose-response and time-course experiments in this context would be valuable to experimentally confirm oxidative stress as a direct driver of PHB1 nuclear trapping. Moreover, the specific SUMOylated nuclear partners mediating PHB1 retention currently remain unidentified. Future proteomic studies are warranted to uncover these key PHB1-bound regulators driving transcriptional repression. ”
Importantly, our mechanistic analysis demonstrates that PHB1 nuclear accumulation is mediated by its SIM-dependent interaction with SUMOylated nuclear proteins, driven by increased UBC9 expression and nuclear SUMO activity in OA. These changes reflect a canonical cellular stress response, particularly in the formation of PML nuclear bodies, which are known to be activated by oxidative stress. Furthermore, the consistent nuclear localization of PHB1 in both human OA cartilage and our in vivo transgenic OA mouse model reinforces this conclusion. Taken together, these findings support a model in which oxidative stress induces SUMOylation pathways, resulting in PHB1 nuclear trapping. We hope this clarifies our rationale and addresses the reviewer’s excellent points.
Reviewer 2 Report
Comments and Suggestions for Authors
This study examines the relationship between PHB1 and SUMOylation in primary osteoarthritis.
Main concerns and comments:
- The title uses "SUMOylation" which is not fit with PHB1 binds to SUMO interacting domain. Will "PHB1 and SUMO interaction" better?
- Figure 1: please explain why PHB1 staining is different from OA groups in (a) and (b)?
- Figure 5: does SUMO2 and SUMO3 SUMOylate PHB1?
- Figure 6: can PHB1 bind UBC9?
- Figure 8: Is "SUMO interaction" better than "SUMOylation" in the figure legend.
- Any UBC9 knockout data to further support the existing data?
Author Response
Comment 1: The title references “SUMOylation,” but PHB1 does not undergo direct SUMOylation. Would “PHB1 and SUMO interaction” be more accurate?
Authors’ Response: We thank the reviewer for this important clarification. We agree that the original title may have unintentionally implied that PHB1 is directly SUMOylated. Since our data indicate that PHB1 accumulates in the nucleus via a SIM-mediated, non-covalent interaction with SUMOylated proteins, we have revised the title to reflect this key mechanism more precisely. Revised Title: “UBC9-Mediated SUMO Pathway Drives Prohibitin-1 Nuclear Accumulation and PITX1 Repression in Primary Osteoarthritis.” We believe this new title more accurately encapsulates our findings.
Comment 2: Figure 1: Please explain why PHB1 staining differs between OA and control groups in panels (a) and (b).
Authors’ Response: Thank you for this observation. The difference in PHB1 staining reflects a fundamental pathological shift in its subcellular localization. In healthy (control) chondrocytes, PHB1 staining is predominantly cytoplasmic, consistent with its well-known mitochondrial role. In contrast, OA chondrocytes exhibit robust nuclear PHB1 accumulation, appearing as distinct nuclear foci. This nuclear re-localization is a hallmark feature of primary OA in our study and is driven by UBC9-mediated activation of the SUMO pathway. We have revised the Figure 1 legend to explicitly describe this subcellular redistribution and its pathological significance. See lines 123-135: ”Figure 1. SUMO-1 proteins accumulate in the nuclei of OA articular chondrocytes and co-localize with PHB1. Representative double immunofluorescence staining against PHB1 (red), SUMO-1 (green) and SUMO-2/3 (green) was carried out on articular chondrocytes of one OA patient and one healthy subject. In OA patients, PHB1 shows a distinct accumulation in the nucleus, in contrast to control cells where it is primarily cytoplasmic. Furthermore, the SUMO proteins accumulate in nuclear bodies (NBs) in OA, while control patient cells show little or no accumulation of SUMOs in the nuclear bodies. (a) In OA chondrocytes, PHB1 (red) is co-localized with SUMO-1 (green), appearing as a strong yellow signal in a merged picture (upper panel). In control cells, PHB1 is largely cytoplasmic with minimal nuclear presence and SUMO-1 nuclear bodies are less prominent (lower panel). (b) In OA chondrocytes, a nuclear accumulation of PHB1 (red) is shown, and is co-localized with SUMO-2/3 (green) (upper panel). In control cells, PHB1 remains predominantly cytoplasmic, and SUMO-2/3 nuclear bodies are less defined (lower panel). ”
Comment 3: Figure 5: Do SUMO-2 and SUMO-3 SUMOylate PHB1?
Authors’ Response: We appreciate this insightful question. Our in vitro SUMOylation assays focused on SUMO-1 and showed no evidence of direct covalent modification of PHB1. Rather, we found that PHB1 associates with SUMO-1 via its SIM, indicating a non-covalent interaction. While SIMs generally recognize both SUMO-1 and SUMO-2/3, some SIMs exhibit paralog specificity. For example, the SIM of DPP9 and DAXX preferentially bind SUMO-1 due to unique structural compatibility (1). Similarly, viral IE1-derived SIMs interact selectively with SUMO-1 but not SUMO-2 (2). Consistent with this concept of specificity, our immunofluorescence data in Figure 1, clearly demonstrate that PHB1 co-localizes with SUMO-1 but not SUMO-2/3 in the nuclei of OA chondrocytes. This supports the notion of a selective interaction between PHB1 and SUMO-1 in the OA context and further validates our mechanistic model.
1.Chang HM, Yeh ETH. SUMO: From Bench to Bedside. Physiol Rev. 2020; 100(4):1599-1619
2. Tripathi V, Chatterjee KS, Das R. Non-covalent Interaction With SUMO Enhances the Activity of Human Cytomegalovirus Protein IE1. Front Cell Dev Biol. 2021; 9:662522
Comment 4: Figure 6: Can PHB1 bind directly to UBC9?
Authors’ Response: This is an excellent question. Our data do not indicate a direct physical interaction between PHB1 and UBC9. Instead, UBC9 enhances the nuclear accumulation of SUMOylated proteins, creating a permissive environment for PHB1’s SIM-mediated recruitment to nuclear compartments. Thus, UBC9 acts indirectly by amplifying the SUMOylation landscape, not by binding PHB1 itself. Clarifications were made to Figure 6 and 8 legends.
Lines 207-221: “Figure 6. PHB1 can bind SUMO1 proteins via a SIM (SUMO-interacting module), which is crucial for its nuclear localization. (a) Diagram represents wild-type PHB1 protein structure and various PHB1 constructs: Wild-type PHB1 (WT PHB1), a mutant where the nuclear signal of export was deleted (PHB1-∆NES) or was replaced by a nuclear localization signal (PHB1-NLS), and a mutant where a putative SUMO-interacting motif (SIM) was deleted (PHB1-∆SIM). All the constructs have a triple Flag-tag at the N-terminal. (b) Co-immunoprecipitation assays with anti-c-Myc antibodies demonstrate that PHB1 interacts with Myc-tagged SUMO1 through the SIM (upper panel). The lower panel indicates the level of Myc-tagged SUMO1 protein in total cell extracts (X-T). (c) The nuclear accumulation of PHB1 is dependent on its SIM. C28/I2, a human chondrocyte cell line, were infected with either flag-tagged wild type (PHB1), (PHB1_NLS), or (PHB1_∆SIM) constructs or empty vector, to produce stable cell lines. The nuclear extract (X-N) and the cytoplasmic extract (X-C) proteins were isolated and analyzed by Western immunoblotting to detect the subcellular presence of flag-tagged PHB1. Anti-GAPDH was used as a cytoplasmic loading control, and anti-Lamin was used as a nuclear loading control. Note the significantly reduced nuclear presence of PHB1-ΔSIM compared to WT PHB1 and PHB1-NLS.”
Lines 247-263: “Figure 8. The UBC9-mediated SUMO pathway stabilizes PHB1 and promotes its nuclear accumulation in U2OS cells. (a) Co-expression of UBC9 and SUMO isoforms enhances PHB1 protein levels. U2OS cells were transfected with the pLPC-3xFlag-PHB1 alone or cotransfected with different components of the SUMOylation pathway (UBC9; UBC9 + Sumo1; UBC9 + Sumo2; UBC9 + Sumo3). Total cell lysates were analyzed by Western blotting using an anti-Flag antibody to detect Flag-PHB1 protein levels. (b) The nuclear accumulation of PHB1 is dependent on its SIM in the presence of UBC9 and SUMO-1. U2OS cells were transfected with Flag-tagged PHB1, PHB1-NLS, or PHB1-ΔSIM constructs, in the presence or absence of co-transfected Myc-SUMO-1 and UBC9. The nuclear proteins (=X-N), as well as total proteins (X-T), were isolated from cells transfected with pLPC-3xFlag-PHB1, PHB1-NLS or PHB1-∆SIM in the presence or absence of myc-SUMO1 and UBC9. Anti-GAPDH was used as a cytoplasmic loading control, and anti-Lamin was used as a nuclear loading control, demonstrating successful cell fractionation. (c) U2OS cells were transfected with UBC9 alone or co-transfected with pCMV4-myc-SUMO1, HA-SUMO2 and myc-SUMO3 or with the empty vector. Total (T), cytoplasmic (C), and nuclear (N) protein extracts were isolated. Western blot analysis using an anti-PHB1 antibody reveals changes in the endogenous PHB1 subcellular localization. Anti-GAPDH and Anti-Lamin A/C were used as loading controls for cytoplasmic and nuclear fractions, respectively.”
Comment 5: For Figure 8, is “SUMO interaction” a more accurate term than “SUMOylation”?
Authors’ Response: We thank the reviewer for this consistent and insightful suggestion. Given that our data clearly show that PHB1 interacts non-covalently with SUMOylated proteins via its SIM, rather than undergoing SUMOylation itself, we agree that “SUMO interaction” is a more accurate term. Accordingly, we have revised the figure legend for Figure 8 to reflect this distinction, see line 247: “Figure 8. The UBC9-mediated SUMO pathway stabilizes PHB1 and promotes its nuclear accumulation in U2OS cells.’’
Comment 6: Are there UBC9 knockout data to further support your findings?
Authors’ Response: This is an important and technically complex issue. Complete knockout of Ube2i (which encodes UBC9) results in early embryonic lethality in mice, with no homozygous embryos surviving beyond embryonic day 9.5. This lethality reflects UBC9’s indispensable role in SUMOylation-dependent nuclear regulation, DNA repair, and cellular viability, effectively precluding the use of global knockout models to study cartilage-specific phenotypes. Although conditional Ube2i knockout models have been successfully developed in other tissues, such as T cells and pancreatic β-cells, no cartilage- or chondrocyte-specific Ube2i conditional knockout models have been reported to date. Conversely, while direct gain-of-function studies of UBC9 in cartilage are currently lacking, there is relevant evidence from related disease models. In a collagen-induced arthritis (CIA) model of rheumatoid arthritis, UBC9 overexpression was shown to enhance synoviocyte proliferation and migration, thereby contributing to joint inflammation and tissue damage (3,4). Notably, UBC9 knockdown in that model attenuated joint destruction, providing indirect but compelling evidence that elevated UBC9 activity can exacerbate joint pathology. Together with findings from our UBC9-overexpressing transgenic mouse model, these data support the notion that increased UBC9 activity may drive disease-like changes in joint tissues. However, no study to date has directly investigated UBC9 gain-of-function specifically in cartilage or chondrocytes, making this a promising and unexplored avenue for future research. We have incorporated this discussion into the revised manuscript to highlight both the current biological constraints and the potential for future targeted investigations in skeletal tissues. See manuscript lines 397-407: “While a UBC9 knockout model would provide valuable loss-of-function data, complete Ube2i knockout results in early embryonic lethality in mice, precluding global knockout studies for cartilage-specific phenotypes. Furthermore, no cartilage- or chondrocyte-specific Ube2i conditional knockout models have been reported. Despite the current lack of direct gain-of-function studies of UBC9 specifically within articular cartilage, indirect support comes from other joint disease models. For instance, in a collagen-induced arthritis model of rheumatoid arthritis, UBC9 overexpression was shown to enhance synoviocyte proliferation and migration, contributing to joint inflammation and tissue damage [64,65]. This evidence, combined with our novel UBC9-overexpressing transgenic mouse model, strongly supports the notion that increased UBC9 activity may broadly drive disease-like changes in joint tissues.”
3. Pap T, Korb-Pap A. Cartilage damage in osteoarthritis and rheumatoid arthritis--two unequal siblings. Nat Rev Rheumatol. 2015; 11(10):606-15.
4. Li, F. et al. SUMO-conjugating enzyme UBC9 promotes proliferation and migration of fibroblast-like synoviocytes in rheumatoid arthritis. Inflammation 37, 1134–1141 (2014).
Round 2
Reviewer 2 Report
Comments and Suggestions for Authors
The authors answered my questions and concerns. No more comments.